# Lineage-level divergence of copepod glycerol transporters and the emergence of isoform-specific trafficking regulation

Marc Catalán-García [1,2], François Chauvigné [2], Jon Anders Stavang[1], Frank Nilsen [1], Joan Cerdà [2✉] & Roderick Nigel Finn [1,2✉]

Transmembrane conductance of small uncharged solutes such as glycerol typically occurs through aquaglyceroporins (Glps), which are commonly encoded by multiple genes in metazoan organisms. To date, however, little is known concerning the evolution of Glps in Crustacea or what forces might underly such apparent gene redundancy. Here, we show that Glp evolution in Crustacea is highly divergent, ranging from single copy genes in species of pedunculate barnacles, tadpole shrimps, isopods, amphipods and decapods to up to 10 copies in diplostracan water fleas although with monophyletic origins in each lineage. By contrast the evolution of Glps in Copepoda appears to be polyphyletic, with surprisingly high rates of gene duplication occurring in a genera- and species-specific manner. Based upon functional experiments on the Glps from a parasitic copepod (*Lepeophtheirus salmonis*), we show that such lineage-level gene duplication and splice variation is coupled with a high rate of neo-functionalization. In the case of *L. salmonis*, splice variation of a given gene resulted in tissue- or sex-specific expression of the channels, with each variant evolving unique sites for protein kinase C (PKC)- or protein kinase A (PKA)-regulation of intracellular membrane trafficking. The combined data sets thus reveal that mutations favouring a high fidelity control of intracellular trafficking regulation can be a selection force for the evolution and retention of multiple Glps in copepods.

[1] Department of Biological Sciences, Bergen High Technology Centre, University of Bergen, Bergen, Norway. [2] IRTA-Institute of Biotechnology and Biomedicine (IBB), Universitat Autònoma de Barcelona, Bellaterra, Cerdanyola del Vallès, Spain. ✉email: joan.cerda@irta.cat; nigel.finn@uib.no

Aquaglyceroporins (Glps) are a phylogenetically distinct grade of water channels (aquaporins) that facilitate the transmembrane conductance of small uncharged solutes such as glycerol, urea or metalloids in addition to water[1–5]. In contrast to the water-selective branches of aquaporins, which typically display narrow selectivity filters composed of four aromatic arignine (ar/R) residues[6–9], the cross-sectional sizes of the Glp selectivity filters are broader and thus facilitate the passage of larger molecules[10–12]. Evolutionary studies have shown that Glps are widespread in both prokrayotic and eukaryotic organisms, but are not ubiquitous, having been lost in certain lineages of protists, plants and insects[2,4,13–15]. In plants and insects, Glps were supplanted by other members of the aquaporin superfamily, either via horizontal gene transfer of nodulin 26-like integral proteins (NIPs) and GlpF-like intrinsic proteins (GIPs) in plants[2,4,13,16,17] or through functional co-option and molecular supplantation by the entomoglyceroporins (Eglps) in hemipteran and holometabolous insects[14,18]. The absence of classical Glps in model organisms such as *Drosophila* has thus obfuscated a deeper understanding of their evolution and function in the arthropod lineage.

Arthropods are a highly diverse, yet monophyletic phylum of joint-legged molting animals that are classified into four major subphyla, the Chelicerata (e.g., sea spiders, horseshoe crabs and arachnids), and a subclade of Mandibulata comprised of the Myriapoda (e.g., centipedes and millipedes), the Crustacea (e.g., waterfleas, tadpole shrimps, barnacles, copepods and decapods) and the Hexapoda (e.g., entognathans and insects)[19]. Glps have been identified in selected species of each subphylum, including multiple genes in arachnids, a single copy in a centipede (*Strigamia maritima*), multiple genes in water fleas and copepods and one or two genes in more basal lineages of Hexapoda[2,14,20–24]. However taxon sampling has remained limited, particularly for Crustacea due to the abscence of genomic and transcriptomic data, and it has thus not been possible to determine whether crustacean Glps are paraphyletic as indicated for Chelicerata or monophyletic as indicated for insects[14,23]. Deciphering the basis for such relationships is an important step toward understanding the origin and divergence of Glp function within each class of organism.

In our previous study, we identified three *glp* genes (*glp1*, −2 and −3) in the parasitic copepod *Lepeophtheirus salmonis*, which expresses the *glp1* and −3 genes as splice variants to form the N-terminal protein isoforms Glp1_v1/v2 and Glp3_v1/v2, respectively[23]. RNA expression profiling revealed that the *glp1_v1* isoform is expressed in pre-adult and adult males, which also occurs in a related species of caligid copepod, *Caligus rogercresseyi*[25,26], while the other transcripts are detected in all stages of the life cycle[23]. In addition, functional analyses of the proteins showed that cAMP was required to promote glycerol transport of the Glp1_v1 isoform[23]. These observations implied that *glp* gene duplication and splice variation may have promoted stage- and sex-specific expression in copepods, and that cAMP-dependent phosphorylation of certain N-terminal residues may be involved in the membrane trafficking regulation of the channels.

To determine whether the molecular regulation of the *L. salmonis* Glps has a common ancestral origin within Copepoda and/or other crustaceans, we used Bayesian inference to re-infer the phylogenetic interrelationships of the *L. salmonis glps* with coding sequences (CDS) assembled from the genomes or transcriptomes of 120 crustaceans including 32 species of copepod together with CDS assembled from myriapod and insect genomes. To further understand whether the derived proteins and/or isoforms have evolved new functions, we developed paralog- and isoform-specific antibodies to determine the cell-type expression sites of

the *L. salmonis* Glps and used site-directed mutagenesis and heterologous expression of the proteins to identify specific residues involved in the Glp intracellular trafficking regulation.

The combined data sets reveal that *glp* gene evolution is highly divergent in Crustacea, ranging from single copies in many species of isopods, amphipods and decapods to up to 10 copies in cladoceran water fleas although with monophyletic origins in each lineage. By contrast the evolution of *glps* in Copepoda appears to be polyphyletic, with surprisingly high rates of gene duplication occurring in a genera- and species-specifc manner. The functional data for the *L. salmonis* Glps further reveal that such lineage-level gene duplication and splice variation can be coupled with a high rate of neofunctionalization. In the case of *L. salmonis*, splice variation of a given gene resulted in tissue- or sex-specific expression of the channels, with each variant evolving unique sites for protein kinase C (PKC)- or protein kinase A (PKA)-regulation of membrane trafficking.

## Results

**A high copy number variation between crustacean Glps.** To reassess the diversity of crustacean Glps in relation to those of other arthropods, we focused on *glp* CDS assembly in the Mandibulata (Fig. 1A), since our previous analysis revealed that mandibulatan *glps* form a separate clade to those of the Chelicerata[14]. Bayesian analysis of an initial codon data set, which included 212 crustacean, four myriapod and 25 insect CDS, revealed that for each lineage, except the Copepoda, *glp* gene evolution appears to be monophyletic with high posterior probabilities (pp) supporting nodes at the level of the order (Fig. 1B; Supplementary Fig. 1A). This is despite high copy number variation between the lineages. For example, within Pancrustacea, single copy genes are found in the majority of taxa from the orders Pedunculata, Notostraca, Isopoda, Amphipoda and Decapoda as well as the class Insecta. Conversely, two paralogs are found in the Euphausiacea, and up to three are encoded in the genomes of Arguloida, Sessilia, and some caridean shrimps and prawns within Decapoda. In the latter case, a separate analysis revealed that the *glp* gene duplication in Caridea may only have occurred within the Palaemonidae family to form the three paralogs (Supplementary Fig. 1B). In addition, *glp* gene copy numbers have rapidly increased in the Daphnidae family of diplostracan water fleas, primarily through tandem duplication. Based upon the gene complement of *D. carinata*, we named the genes A1–A9 and B1–B3 in accordance with their phylogenetic distributions and their genomic loci in two linkage groups (Supplementary Fig. 1C). Although multiple *glps* were also detected in Myriapoda, they form a separate clade to those of the Pancrustacea, and consequently *glp* gene evolution in the Myriapoda initially appeared to be monophyletic. However, a separate analysis of *glp* CDS from 28 species of myriapods, indicates that within the classes Pauropoda, Symphyla and Diplopoda, *glp* evolution is polyphyletic with genes separated into two potential subclades A and B (Supplementary Fig. 1D).

In contrast to the monophylic gene families observed in most Crustacea, *glp* gene evolution in the Copepoda appears to be polyphyletic (Fig. 1B) with one clade of genes clustering as a sister branch to the Thecostraca and Arguloida (pp = 0.85), and a second clade clustering between Insecta and Branchiopoda (pp = 0.5). Hence, although nodes at the level of organismal order were well supported with pp > 0.95, statistical support values between the classes of organisms was low, indicating incomplete taxon sampling.

To further investigate the evolution of *glp* genes within Copepoda, we assembled 98 *glp* CDS from 32 species within 16 families from four orders (Siphonostomatoida, Cyclopoida,

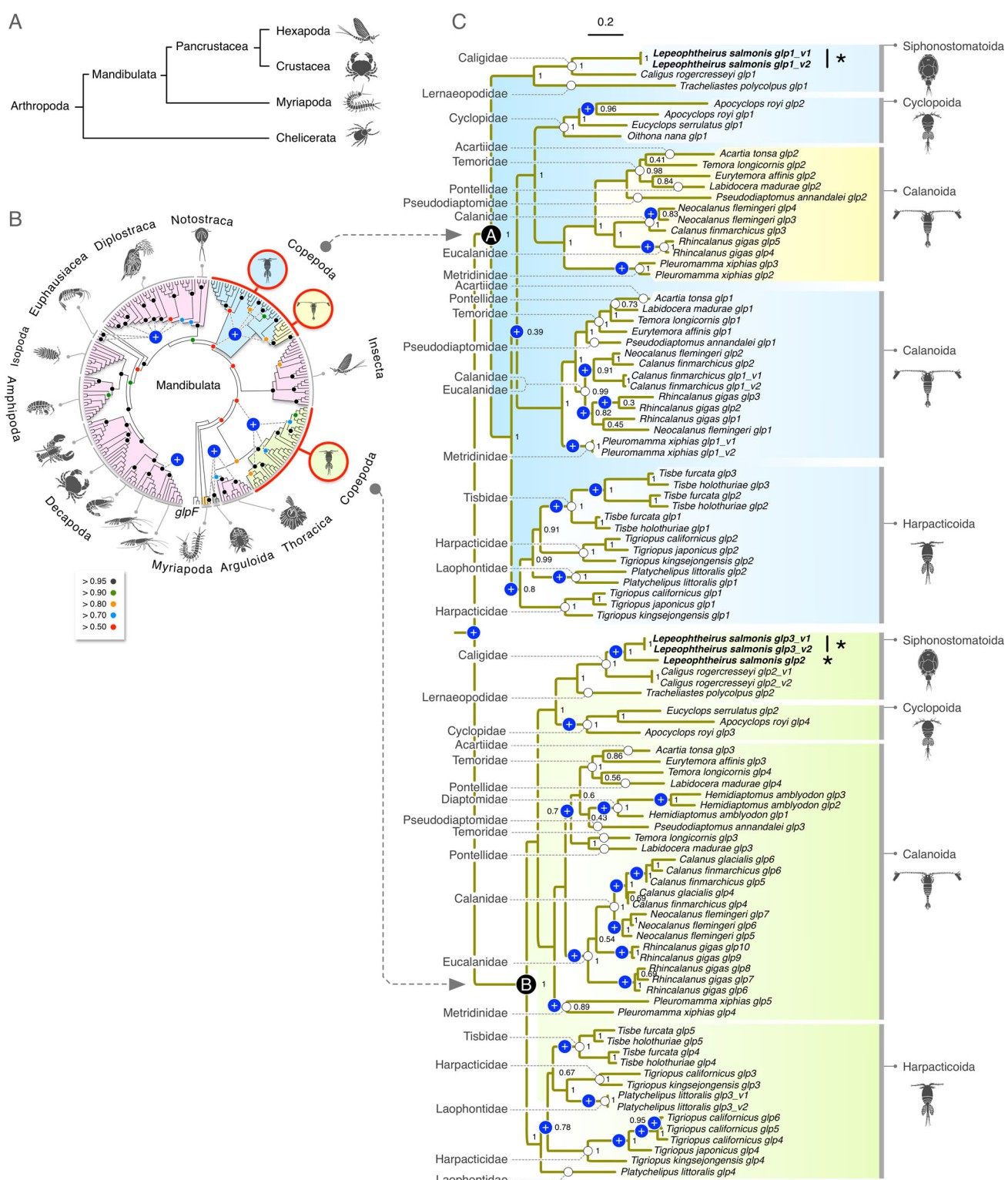

**Fig. 1 Molecular phylogeny of glycerol transporters in Mandibulata. A** Phylogenetic interrelationships of extant arthropod subphyla after Gribet and Edgecombe[19]. **B** Bayesian majority rule consensus tree (15 million MCMC generations) of 224,094 nucleotide sites of 249 mandibulaten CDS partitioned by codon. The tree is rooted with eubacterial *glpF*. Bayesian posterior probabilities as indicated by the colored dots in the key are annotated on selected nodes. The scale bar indicates the expected rate of substitutions per site. Major incidences of gene duplications are indicated by a white "+" on a blue background. (See Supplementary Fig. 1 for the fully annotated). **C** Bayesian majority rule consensus tree (5 million MCMC generations) of 85,152 nucleotide sites of 98 copepod CDS partitioned by codon. The tree is midpoint rooted and annotated with the respective copepod families on the left and orders on the right. Posterior probabilities are annotated at each node with the scale bar indicating the expected rate of substitutions per site. Gene duplications are indicated by a white "+" on a blue background.

Calanoida and Harpacticoida), and computed their interrelationshps via Bayesian inference. The resultant tree generated the two clades identified in the crustacean analysis with high statistical support (pp = 1.0) and we therefore named them A and B (Fig. 1C). The tree topology reveals that although *glp* gene copy numbers vary considerably, gene origins within each clade are probably monophyletic for the Siphonostomatoida, Cyclopoida and Harpacticoida orders, but polyphyletic for the Calanoida order within the A clade. A surprising feature revealed by the analysis is nevertheless the high level of gene duplication in the different lineages. With the exception of the ancestral gene duplication that gave rise to the A and B clades, we detected 31 duplications within the four copepod orders and the presence of splice-variants within each of the orders except Cyclopoida (Fig. 1C; Supplementary Fig. 1A). The data thus show that *glp* gene expansions within Copepoda are mostly occurring in a family- and genera-specific manner, and that the *L. salmonis glp* gene and isoform complement thus evolved specifically within the genus. Indeed, although gene duplicates and splice variants also exist in the closely related caligid copepod *Caligus rogercresseyi*, there appear to be only two genes, rather than three[26] and the *glp* repertoires are thus not fully conserved between the genera.

**Expression and cellular localization of *L. salmonis* Glps.** To investigate the expression and cellular localization of the *L. salmonis* Glps, we produced affinity-purified antibodies against each paralog and isoform. The specificity of each antibody was tested by Western blot analysis on total membrane protein extracts from *Xenopus laevis* oocytes expressing the full complement of the *L. salmonis* aquaporins (Bib, PripL, Aqp12L1; Aqp12L2, Glp1_v1; Glp1_v2, Glp2, Glp3_v1 and Glp3_v2; Fig. 2A–E, Supplementary Fig. S2A–E). The results show that each of the *L. salmonis* Glp antisera generated specifically recognized its corresponding antigen, therefore likely indicating that these antibodies do not cross-react with any of the other aquaporins. For some channel variants, including Glp1_v1, Glp1_v2 and Glp3_v1, bands of ~18–22 kDa were detected, indicating some degradation in the *X. laevis* oocyte expression system (Fig. 2A, B, D). A higher band of ~60 kDa was detected with the Glp3_v2 antibody indicating that the Glp3_v2 isoform is also present as a dimer (Fig. 2E).

Western blots for Glp1_v1, Glp1_v2, Glp2, Glp3_v1 and Glp3_v2 on protein extracts from adult male and female *L. salmonis* showed strongly reactive bands for each of the antibodies of approximately the same molecular mass as the predicted monomers (Glp1_v1: 34.4; Glp1_v2: 32.4; Glp2: 28.6;

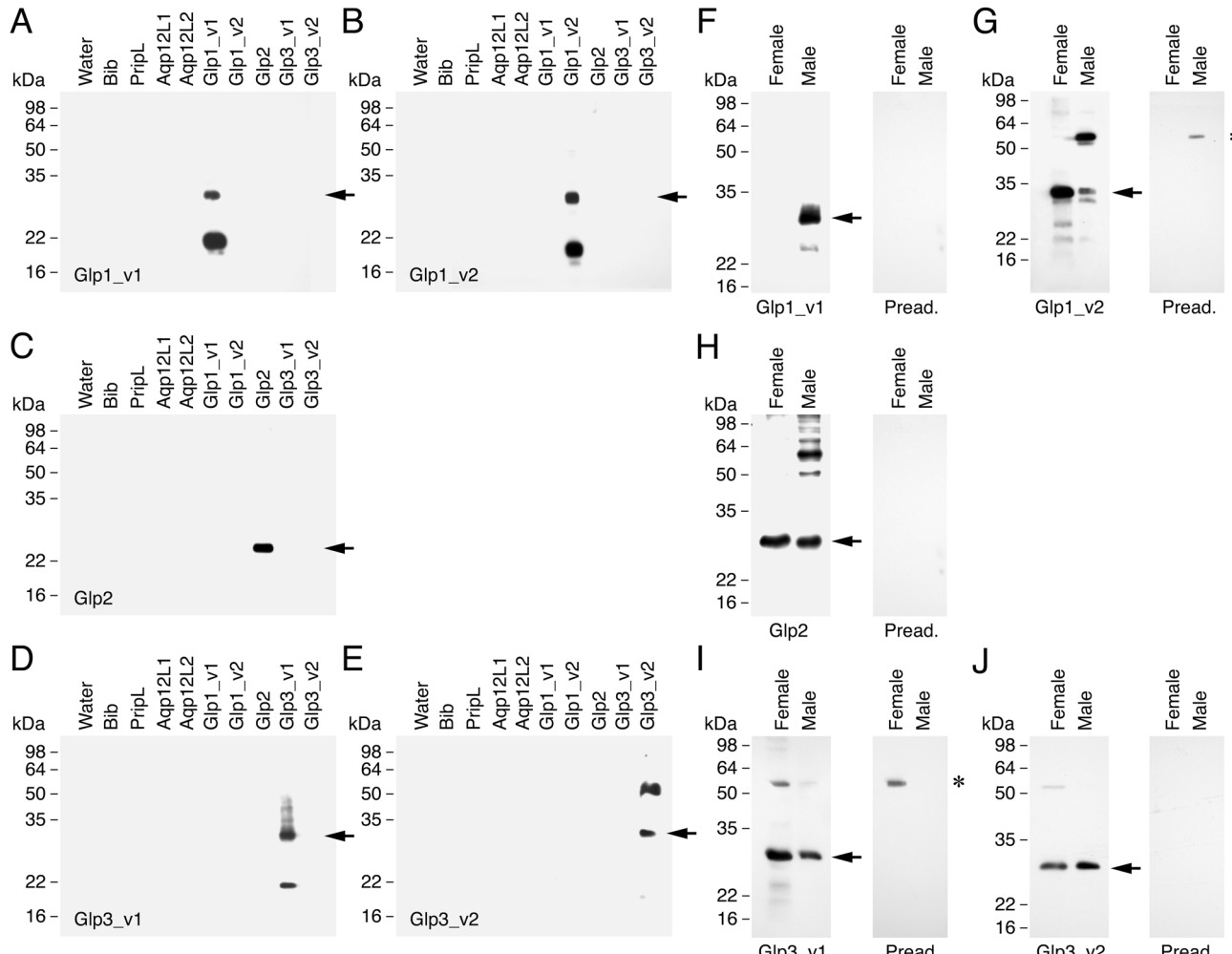

**Fig. 2 Antibody specificity against *L. salmonis* Glps. A–E** Western blot of total membranes of *X. laevis* oocytes injected with water or expressing different *L. salmonis* aquaporins. Membranes were probed with paralog-specific antibodies against Glp1_v1, Glp1_v2, Glp2, Glp3_v1 or Glp3_v2 as indicated. Note that none of the antisera showed cross-reactivity with another aquaporin. **F–J** Detection of Glps in protein extracts from adult whole female and male *L. salmonis*. The blot on the right in each panel was incubated with the corresponding primary antibody preadsorbed with the antigenic peptide. The asterisks in G and I indicate the cross-reaction of the Glp1_v2 and Glp3_v1 antisera, respectively, with a polypeptide of ~55 kDa in males or females. In all panels, the aquaporin monomers are indicated with an arrow, whereas molecular mass markers (kDa) are on the left.

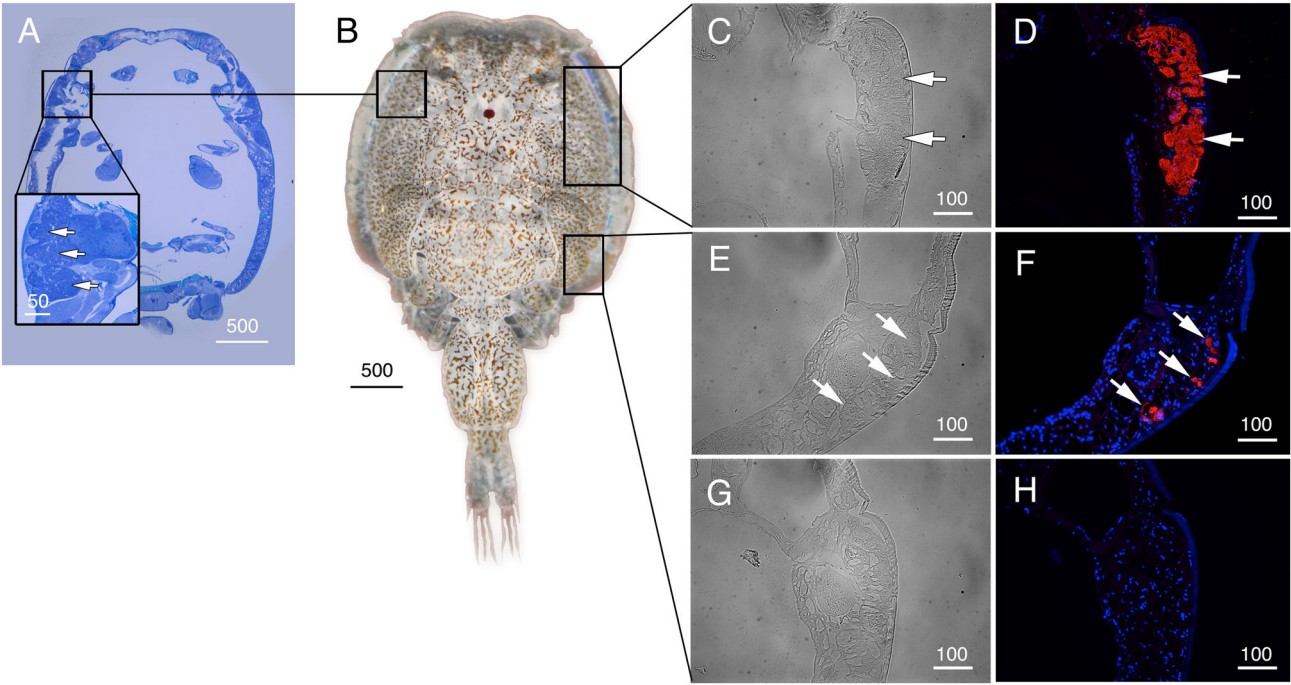

**Fig. 3 Localization of Glp1_v1 in male *L. salmonis*.** Representative histological section of adult male **A** *L. salmonis* stained with toluidine blue **B** showing the proximate location of the type 3 tegumental glands (inset). Bright field (**C**, **E** and **G**) and immunofluorescence (**D**, **F**) microscopy images of Glp1_v1 localization (arrows). Sections were labeled with affinity-purified *L. salmonis* Glp1_v1 antiserum (red) and counterstained with 4′,6-diamidino-2-phenylindole (DAPI; blue). **H** Control sections incubated with preadsorbed antiserum were negative. Scale bars are in μm.

Glp3_v1: 34.5; Glp3_v2: 35.8 kDa) (Fig. 2A–E). Antibodies that were preadsorbed by the antigenic peptides were negative for the same bands (Fig. 2F–J, Supplementary Fig. S2F–O). These data reveal that Glp1_v1 is specific to adults males (Fig. 2F) as previous noted for the RNA[23], while Glp1_v2, Glp2, Glp3_v1 and Glp3_v2 are expressed in both sexes (Fig. 2 G–J). In the case of Glp1_v2, however, a more intense reaction in the female compared to the male could be related to an additonal sex-specific expression in the oocytes (see below).

The cellular localizations of the Glps in adult male and female *L. salmonis* were subsequently determined by immunofluorescence microscopy using the affinity-purified antibodies. The male-specifc expression of Glp1_v1 was detected in type 3 tegumental glands that are bilaterally located in the subepidermal tissue of the cephalothorax (Fig. 3A–F). The glands specifically develop in preadults and adults and extend posterolaterally along the cephalothorax[27]. The long duct structures, which discharge secretions far from the main gland tissues, are indicated by the smaller area of positive staining as the gland extends poster-olaterrally (Fig. 3D, F). Preadsorption of the antiserum with the peptide antigen led to a complete absence of staining in the same tissues (Fig. 3G, H). In contrast to Glp1_v1, the Glp1_v2 isoform is expressed in the enterocytes throughout the lengths of the intestines of both males (Fig. 4A–E) and females (Fig. 4F, Supplementary Fig. S3A, B). In these experiments, the channel was localized intracellularly, and preadsorbed antibodies yielded no signal (Supplementary Fig. S3C–F; K–L). In females, Glp1_v2 was also localized intacellulary within ovarian oocytes (Fig. 4G, H) and the cortical cytoplasm of the stacked oocytes in the immature egg strings (Fig. 4I, J). Preadsorbed antibodies gave no signal in the oocytes (Supplementary Fig. S3G, H). Glp2 was found in the epithelia lining type 1 tegumental glands, which are numerous in both males (Fig. 5A–C) and females (Fig. 5D–F). No signal was detected by preadsorbed antibodies (Fig. 5G, H). Glp3_v1 is specifically located in the apical membrane

throughout the intestinal brush border of both males (Fig. 6A–C) and females (Fig. 6F–J), with no signal detected by preadsorbed antibodies (Fig. 6D, E; Supplementary Fig. S3I, J). Finally Glp3_v2 is localized in the the epithelia of blood vessels and blood sinuses surrounding the intestines of both males (Fig. 7A–C) and females (Fig. 7F–H) with no staining observed when using preadsorbed antibodies (Fig 7D, E, I, J). These data thus show that the closely related Glp2 and Glp3 paralogs, including both splice variants of the Glp3 channel are each expressed in different tissues. Conversely, isoforms of the Glp1 and Glp3 paralogs (Glp1_v2 and Glp3_v1), which are distantly related in the A and B tree clusters, are both expressed in the enterocytes, but with differential subcellular localization. These observations reveal that the signal transduction pathways regulating the intracellular channel trafficking of Glp1_v2 and Glp3_v1 in the enterocytes may be different.

**Protein kinases regulate the intracellular trafficking of *L. salmonis* Glps**. To investigate whether PKC or PKA signal trans-duction pathways are involved in the intracellular trafficking of the *L. salmonis* Glps, we expressed each paralog and variant in *X. laevis* oocytes exposed to either the PKC activator phorbol 12-myristate 13-acetate (PMA) or the cAMP-PKA activator, for-skolin (FSK), respectively. In the latter instance, the FSK-exposed oocytes were preincubated with the phosphodiesterase inhibitor 3-isobutyl-1-methylxanthine (IBMX). We then visualized the changes in plasma membrane channel content via immuno-fluorescence and image analysis of the frog oocytes, as well as by Western blots of the total and plasma membrane extracts using the isoform-specific antibodies (Fig. 8; Supplementary Data 2). Injection of Glp1_v1 resulted in significant increases ($p < 0.001$; one-way ANOVA with Dunnett's multiple comparison test) of the channel in the plasma membrane fraction compared to the controls controls (treated with the DMSO vehicle) when exposed to PMA or FSK (Fig. 8A–C; Supplementary Fig. S4A–B).

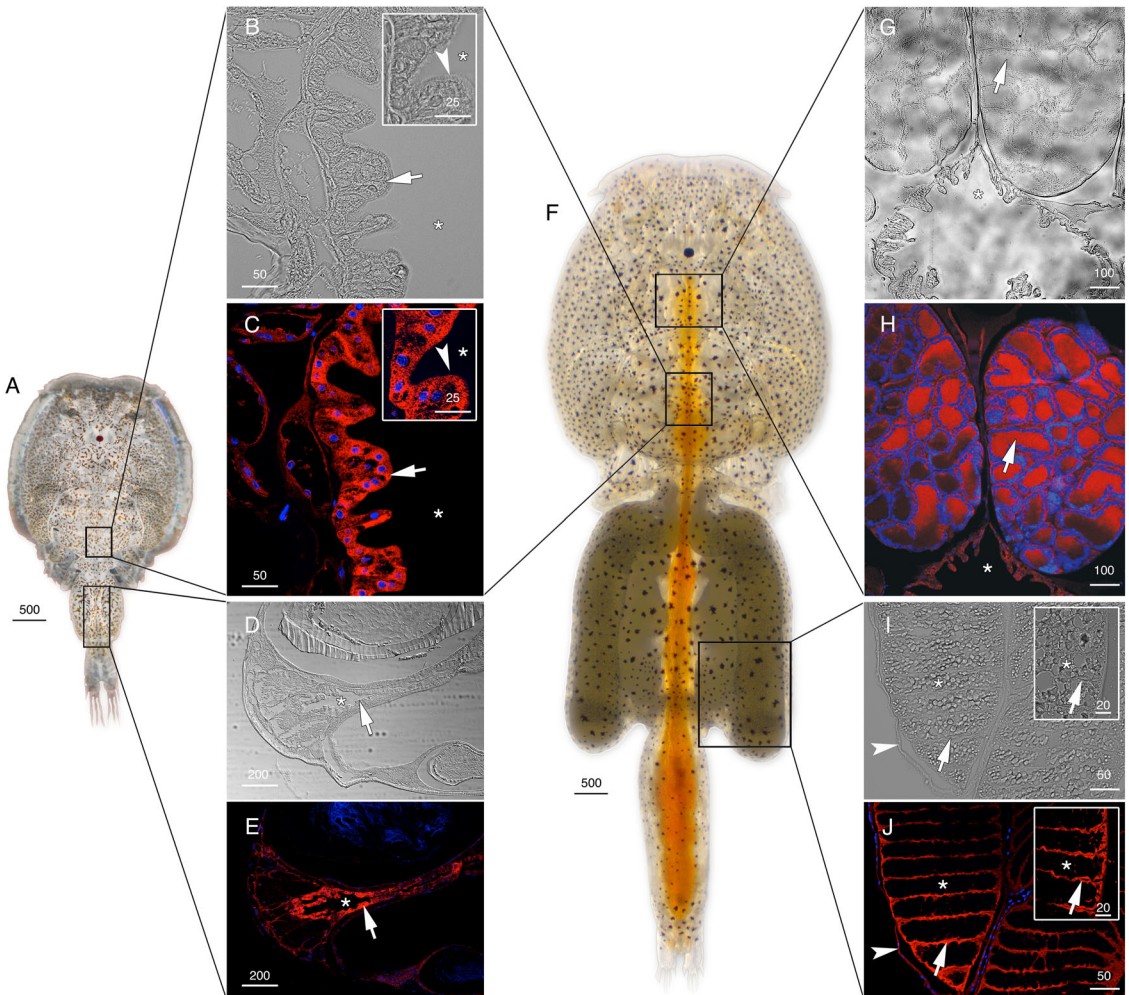

**Fig. 4 Localization of Glp1_v2 in *L. salmonis*.** Adult male (**A**) and female (**F**) representative bright field (**B**, **D**, **G** and **I**) and immunofluorescence (**C**, **E**, **H** and **J**) microscopy images of Glp1_v2 localization (arrows). Sections were labeled with affinity-purified *L. salmonis* Glp1_v2 antiserum (red) and counterstained with DAPI (blue). Asterisks indicate intestinal lumen (**D**, **C**, **D**, **E**, **G** and **H**) or oocyte (**I** and **J**), and arrowheads indicate the brush border (inset **B** and **C**) or epithelia lining the immature egg strings (**I** and **J**). Scale bars are in μm.

Conversely, the experiments with Glp1_v2 (Fig. 8D–F: Supplementary Fig. S4C–D), Glp2 (Fig. 8G–I; Supplementary Fig. S4E–F) and Glp3_v1 (Fig. 8J–L; Supplementary Fig. S4G–H) only resulted in significant increases ($p < 0.001$; one-way ANOVA with Dunnett's multiple comparison test) of the channels in the plasma membrane fractions when the frog oocytes were exposed to FSK. However, in oocytes expressing Glp3_v2, no changes in the amount of the channel in the plasma membrane were observed after PMA or FSK treatments (Fig. 8M–O; Supplementary Fig. S4I–J).

To corroborate the above results we investigated the changes in osmotic water ($P_f$) and glycerol ($P_{gly}$) permeability of oocytes expressing each paralog and variant under the same experimental conditions (Fig. 9; Supplementary Data 2). These data show that the $P_f$ and $P_{gly}$ of Glp1_v1-injected oocytes were significantly increased ($p < 0.05$; one-way ANOVA with Dunnett's multiple comparison test) compared to controls in the presence of PMA or FSK (Fig. 9A). In fact, exposure of the oocytes to either activator was required in order to detect the increase in $P_f$ or $P_{gly}$ with respect to water-injected controls. This was also the case for the Glp1_v2 isoform, although in this latter case, a significant increase ($p < 0.05$; one-way ANOVA with Dunnett's multiple comparison test) in $P_f$ and $P_{gly}$ was only detected in the presence of FSK (Fig. 9B). The $P_f$ and $P_{gly}$ of Glp2 and Glp3_v1-injected

oocytes also increased significantly ($p < 0.05$; one-way ANOVA with Dunnett's multiple comparison test) with respect to controls in the presence of FSK (Fig. 9C–D). As in the previous immunofluoresence experiments, no changes in the $P_f$ or $P_{gly}$ of Glp3_v2-injected oocyte were detected in relation to the controls when the oocytes were exposed to either PMA or FSK (Fig. 9E). Taken together these independent experiments reveal that the intracellular trafficking of the Glp1_v1 isoform is activated by both the PKC and PKA signal transduction pathways, yet only the PKA and not the PKC signal transduction pathway is involved in the intracellular trafficking induction of the Glp1_v2, Glp2 and Glp3_v1 channels in oocytes. Conversely, neither the PKC nor the PKA signal transduction pathways appear to regulate the intracellular trafficking of the Glp3_v2 isoform.

**Isoform-specific sites regulate the membrane trafficking of *L. salmonis* Glps.** To test the hypothesis that intracellular trafficking of Glp1_v1, Glp1_v2, Glp2 and Glp3_v1 channels is controlled by PKC and/or PKA phosphorylation of the channels, we initially conducted *in silico* searches for relevant phosphorylation sites in the intracellular domains of each channel. This yielded several potential sites in the N-termini or loop B (Fig. 10A). To determine whether such sites are functional for either of the kinases,

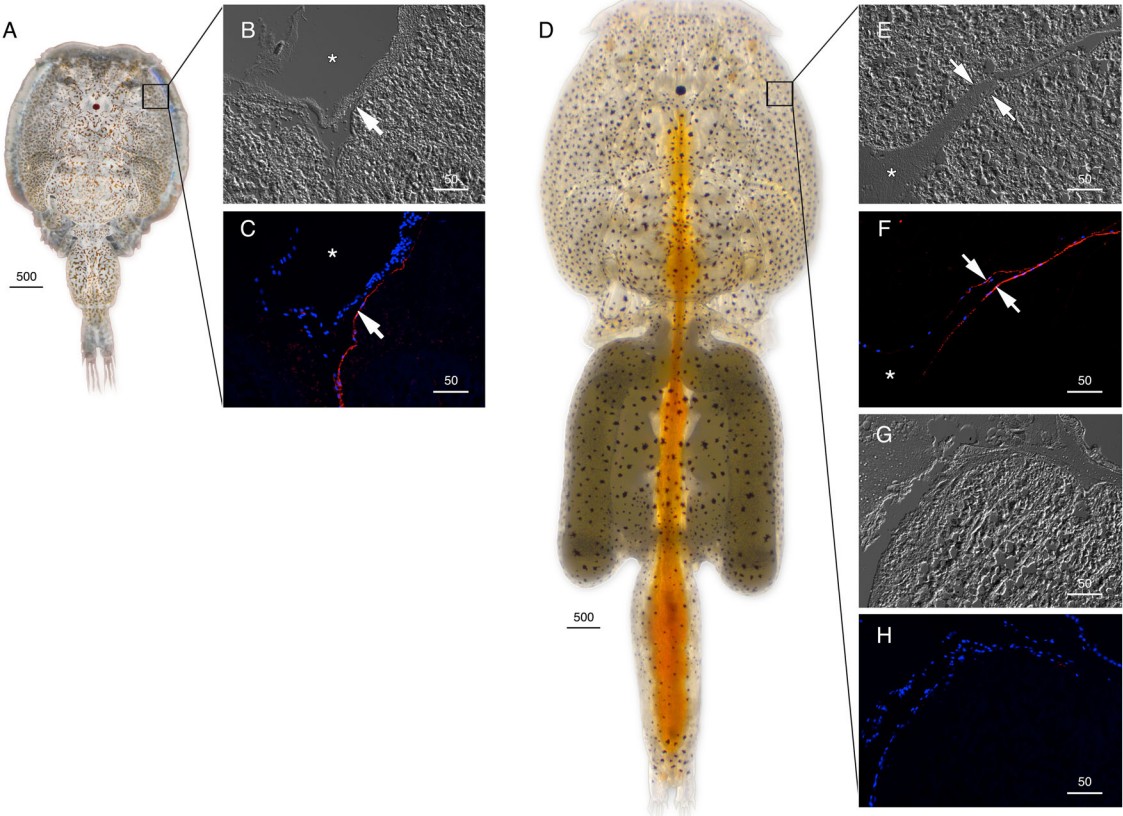

**Fig. 5 Localization of Glp2 in *L. salmonis*.** Adult male (**A**) and female (**D**) representative bright field (**B**, **E** and **G**) and immunofluorescence (**C**, **F** and **H**) microscopy images of Glp2 localization in epithelia (arrows) of type 1 tegumental glands. Sections were labeled with affinity-purified *L. salmonis* Glp2 antiserum (red) and counterstained with DAPI (blue). Control sections (**H**) incubated with preadsorbed antiserum were negative. Asterisks indicate coelum. Scale bars are in μm.

we mutated each to an aspartate (D), which mimics constitutive phosphorylation, and re-examined the $P_f$ of oocytes expressing each mutant under exposure to DMSO, PMA or FSK as above. The equivalent expression of each mutant in relation to the wild type was validated via Western blots of total membrane protein extracts using the isoform-specific antibodies.

For oocytes expressing wild-type Glp1_v1, the $P_f$ was further elevated in response to PMA or FSK, as observed previously, while the $P_f$ of the Glp1_v1-T3D mutant oocytes was not increased with PMA, but remained stimulated by FSK (Fig. 10B; Supplementary Data 2). This reveals that Glp1_v1 T3 is a functional PKC site, but that another PKA site appears to exist. In contrast, oocytes expressing the Glp1_v1-T14D mutant showed the same changes in $P_f$ after PMA or FSK treatment as those expressing the wild-type, whereas the $P_f$ of the Glp1_v1-S111D oocytes was positively affected by PMA but not by FSK (Fig. 10B). We therefore concluded that T3 and S111 are the functional PKC and PKA sites in Glp1_v1, respectively.

For Glp1_v2, oocytes expressing the Glp1_v2-S6D mutant mimmicked the effect of PMA and FSK on the wild-type (Fig. 10C; Supplementary Data 2). However the Glp1_v2-S94D mutant oocytes showed a constitutively elevated $P_f$ with respect to the wild-type, while FSK had no effect (Fig. 10C). Consequently, S94 appears to be the functional PKA site in Glp1_v2.

For Glp2, only one potential PKA site (S10) was identified in the N-terminus of the channel. Oocytes expressing the Glp2-S10D mutant showed an increased $P_f$ with respect to that of the wild-type, while the positive effect of FSK observed in the wild-type was abolished in the mutant (Fig. 10E; Supplementary

Data 2). This demonstrates that S10 is the PKA functional site in Glp2.

Finally, of the two potential PKA sites (S5 and S43) found in the N-terminus of Glp3_v1, only the oocytes expressing the Glp3_v1-S5D mutant showed an elevated $P_f$ with respect to the wild-type, which was not affected further by FSK (Fig. 10F; Supplementary Data 2). Conversely, the oocytes expressing the Glp3_v1-S43D mutant mimicked the effect of FSK on the wild-type (Fig. 10F), revealing that S5 and not S43 is the functional PKA site.

For all of the paralogs and their isoforms, immunoblotting experiments showed that oocytes expressed equivalent amounts of the wild-type and mutants (Fig. 10D and G; Supplementary Fig. 5A–D), indicating that the observed effects were not caused by differential expression mechanisms. These experiments thus confirm that the intracellular trafficking of four of the five *L. salmonis* Glp channels can be induced by the PKC and/or PKA signal transduction pathways.

## Discussion

The present phylogenetic analysis of *glp* CDS in Crustacea is to the best of our knowledge the first to reveal the striking variability in *glp* gene copy number between the different lineages. It is surprising to find only single-copy *glps* in the the species of pedunculate thecostracans (barnacles), as well as the majority of species of isopods, amphipods and decapods investigated. This contrasts the moderate to high levels of *glp* gene redundancy in other crustacean lineages, such as the euphausiid krills with two

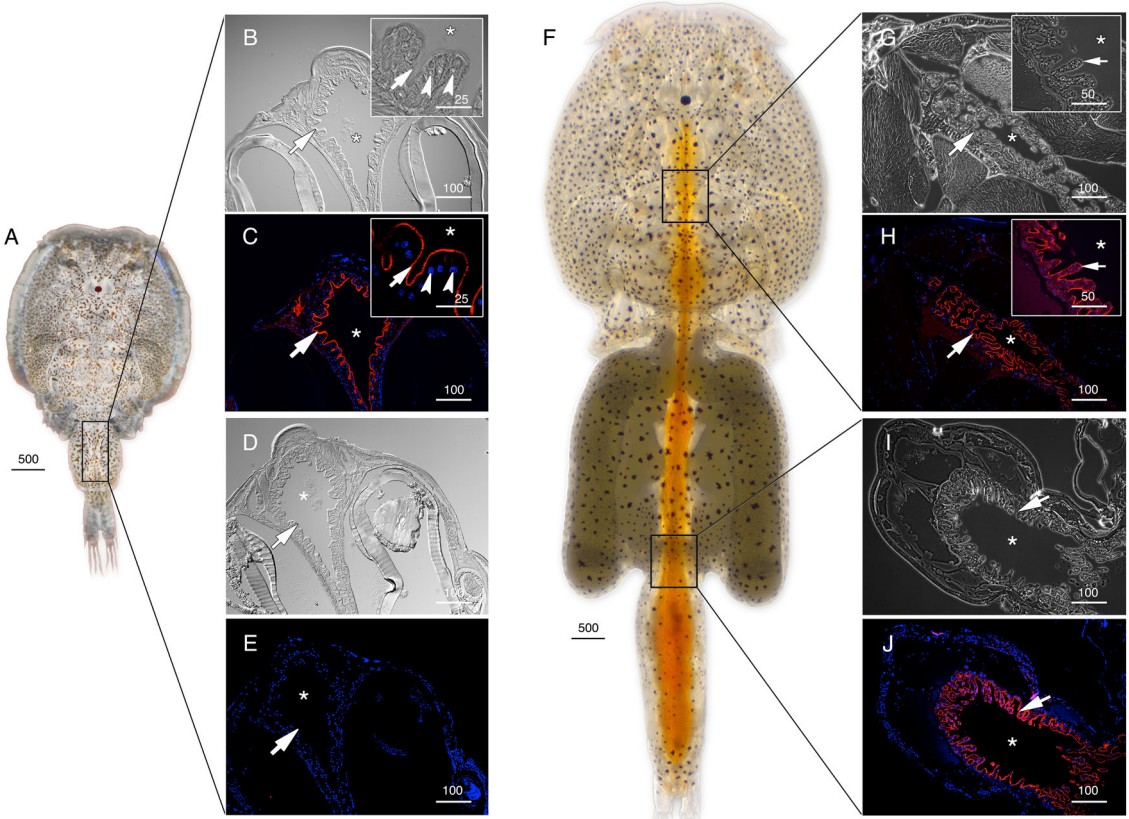

**Fig. 6 Localization of Glp3_v1 in *L. salmonis*.** Adult male (**A**) and female (**F**) representative bright field (**B**, **D**, **G** and **I**) and immunofluorescence (**C**, **E**, **H** and **J**) microscopy images of Glp3_v1 localization in the brush border (arrows). Sections were labeled with affinity-purified *L. salmonis* Glp3_v1 antiserum (red) and counterstained with DAPI (blue). Insets in **B**, **C**, **G** and **H** are higher magnifications. Control sections (**E**) incubated with preadsorbed antiserum were negative. Asterisks indicate intestinal lumen, and arrowheads (insets **B** and **C**) indicate nuclei of individual enterocytes. Scale bars are in µm.

copies, the sessilian thecostracans and palaemonidan prawns with up to three copies, and the diplostracan water fleas and calanoid copepods with up to nine or ten paralogs in a given species. These latter levels of *glp* gene redundancy have, however, been reported in other lineages of Ecdysozoa. For example, Nematoda and Tardigrada encode between five to eight *glp* paralogs in their genomes, whereas diverse lineages of Chelicerata, including arachnid ticks, scorpions and spiders evolved between three to five *glp* paralogs[14,28,29]. A review of the *glp* gene complement in a more distantly related chelicerate, the Atlantic horseshoe crab (*Limulus polyphemus*) also reveals eight paralogs encoded in the genome of this species. As previously reported for vertebrates[2,15,30], the basis for some of the higher *glp* copy numbers in chelicerates such as *L. polyphemus* and the arachnids is partially rooted in ancestral whole genome duplications (WGD)[31,32]. However, although polyploidy is recognized in the isopod *Trichonicus* sp., the amphipod *Pontoporeia affinis*, and a parthenogenic strain of the anacostracan brine shrimp (*Artemia* sp.)[33], WGD is not widely known to have occurred in Crustacea despite large variations in the sizes of their genomes[34,35]. Consequently other mechanisms of gene duplication must have generated the *glp* redundancy in this lineage.

The analysis of the *glp* complements in the diplostracan water fleas revealed a high level of gene linkage in the two species *D. carinata* and *D. pulex* with sequences assigned to chromosomes. Although the syntenic relationships are not fully conserved due to block rearrangements between the species (see Supplementary Fig. 1B), it seems likely that tandem duplication was a major driver of *glp* expansion in this lineage. This is consistent with the

high prevalence of tandem gene clusters in the genomes[36,37]. Conversely, the increased repertoire of *glp* channels in the palaemonidan prawns seems to be associated with the burst of transposon activity that shaped their genomes[38]. This latter mechanism is also thought to have shaped the very large genomes of the euphausiid krills, which are between ~4 – 14 times longer than the human genome[34]. Hence tandem duplication and transposon activity can explain the increases in *glp* copy numbers in several lineages of Crustacea. Since such redundancy is thought to buffer phenotypes from genomic variations and thus confer advantages for an organism´s ability to evolve[39], it is surprising to note the lack of *glp* redundancy in so many species.

This is not the case for the Copepoda. In contrast to the other lineages of Crustacea studied, the copepod *glps* appear to have polyphyletic origins. This would also imply asymmetrical gene loss in the other lineages of Crustacea. If long-branch attraction is discarded, then the lack of *glp* redundancy may be due to asymmetric loss of the genes that are orthologous to the two copepod clusters. Such gene losses are suggested to have occurred on a large-scale in the Insecta, which tend to have single-copy *glps* or have lost them completely[14], yet are considered to have experienced multiple rounds of WGD during their evolution[40]. In the present analysis, however, data were only available for four orders of Copepoda, and it is thus too early to draw conclusions on the definitive origins of the copepod *glps*. What is clear, however, is that within the four orders of Copepoda analyzed, there are broad levels of species- and genera-specific duplications of the *glps*. In addition, within three of the orders, Calanoida, Harpacticoida and Siphonostomatoida, we also found evidence

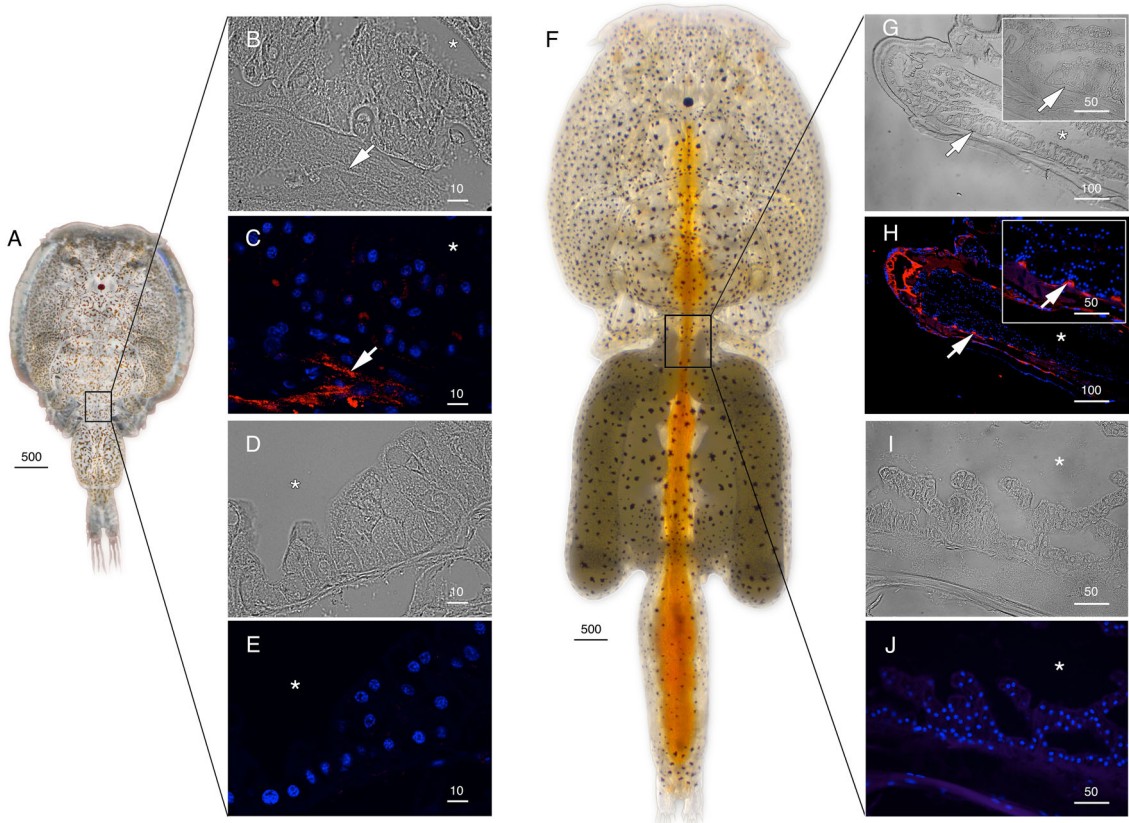

**Fig. 7 Localization of Glp3_v2 in *L. salmonis*.** Adult male (**A**) and female (**F**) representative bright field (**B**, **D**, **G** and **I**) and immunofluorescence (**C**, **E**, **H** and **J**) microscopy images of Glp3_v2 localization (arrows) surrounding the intestine. Sections were labeled with affinity-purified *L. salmonis* Glp3_v2 antiserum (red) and counterstained with DAPI (blue). Control sections (**E** and **J**) incubated with preadsorbed antiserum were negative. Asterisks indicate intestinal lumen, and insets in **G** and **H** show higher magnifications. Scale bars are in μm.

that the functional repertoires are further increased through splice variation.

To gain insight into the molecular basis for the retention and function of the different Glp paralogs and isoforms in copepods, we investigated the cellular localization of each variant in the hematophagous *L. salmonis* copepod. We selected this model, since we had previously shown that the transcripts are expressed in the adults and each translated protein functions as a Glp[23]. The immunolocalization data revealed that the *L. salmonis* Glps are expressed in five different tissues, the type 3 tegumental glands of males (Glp1_v1), and the type 1 tegumental glands (Glp2), the blood vessels and blood sinuses surrounding the intestine (Glp3_v2), and the enterocytes (Glp1_v2, Glp3_v1) of both sexes. In addition, the Glp1_v2 splice variant is expressed in the oocytes of females. Such divergent tissue localizations indicate that the Glps have in most instances neofunctionalized rather than sub-functionalised to play specific roles in the fluid and nutrient homeostasis of *L. salmonis*. However, in the case of Glp1_v2 and Glp3_v1 variants, which derive from the distantly related A and B clusters, respectively, there is redundant expression in the same enterocytes, but they are not colocated in the plasma membrane. Indeed the intracellular localization of Glp1_v2 in the enterocytes was also seen in the female oocytes in a pattern that is highly reminiscent of Aqp1ab in marine teleost oocytes[41–43]. The respective intracellular and apical location of the Glp1_v2 and Glp3_v1 variants in the enterocytes might therefore represent a form of subfunctionalization.

These observations promted us to investigate the intracellular trafficking regulation of the different Glp channels from *L.*

*salmonis*. Since reversible phosphorylation of specific amino acid sites induced by vasopressin- or vasotocin-related neuropeptides activating PKA and PKC signal transduction pathways is a well-established mechanism governing the intracellular trafficking of aquaporins[44–46], we initially tested whether such pathways can regulate the *L. salmonis* Glps. Independent experiments that examined the fractional change in plasma membrane content and the $P_f$ of *X. laevis* oocytes expressing the Glps when exposed to PMA and FSK provided consistent evidence that the PKC and PKA pathways activate the membrane insertion of Glp1_v1, while plasma membrane trafficking of Glp1_v2, Glp2 and Glp3_v1 is regulated by PKA only. In contrast, neither of these two pathways regulate the intracellar trafficking of Glp3_v2. The most direct evidence was obtained from the site-directed mutagenesis experiments, which demonstrated that phosphorylation of specific channel residues by PKC or PKA indeed differentially regulates the intracellular trafficking of the *L. salmonis* Glps. However, the data futher revealed that not all of the predicted sites are functional, which precludes definitive comparisons with the Glps of other species based solely upon *in silico* predictions. The data for *L. salmonis* nevertheless reveal that there is isoform-specific pathway regulation of the channels with PKC and PKA regulating the Glp1_v1 variant, but only PKA regulating the Glp1_v2 variant. Conversely, the PKA pathway regulates the Glp3_v1 variant, but not the Glp3_v2 variant. As a result, the PKA pathway can regulate four channels that are each expressed in different tissues, with extra controls added by the PKC pathway for the male-specifc Glp1_v1 isoform. Intriguingly, we found that the two Glp paralogs expressed in enterocytes (Glp1_v2 and

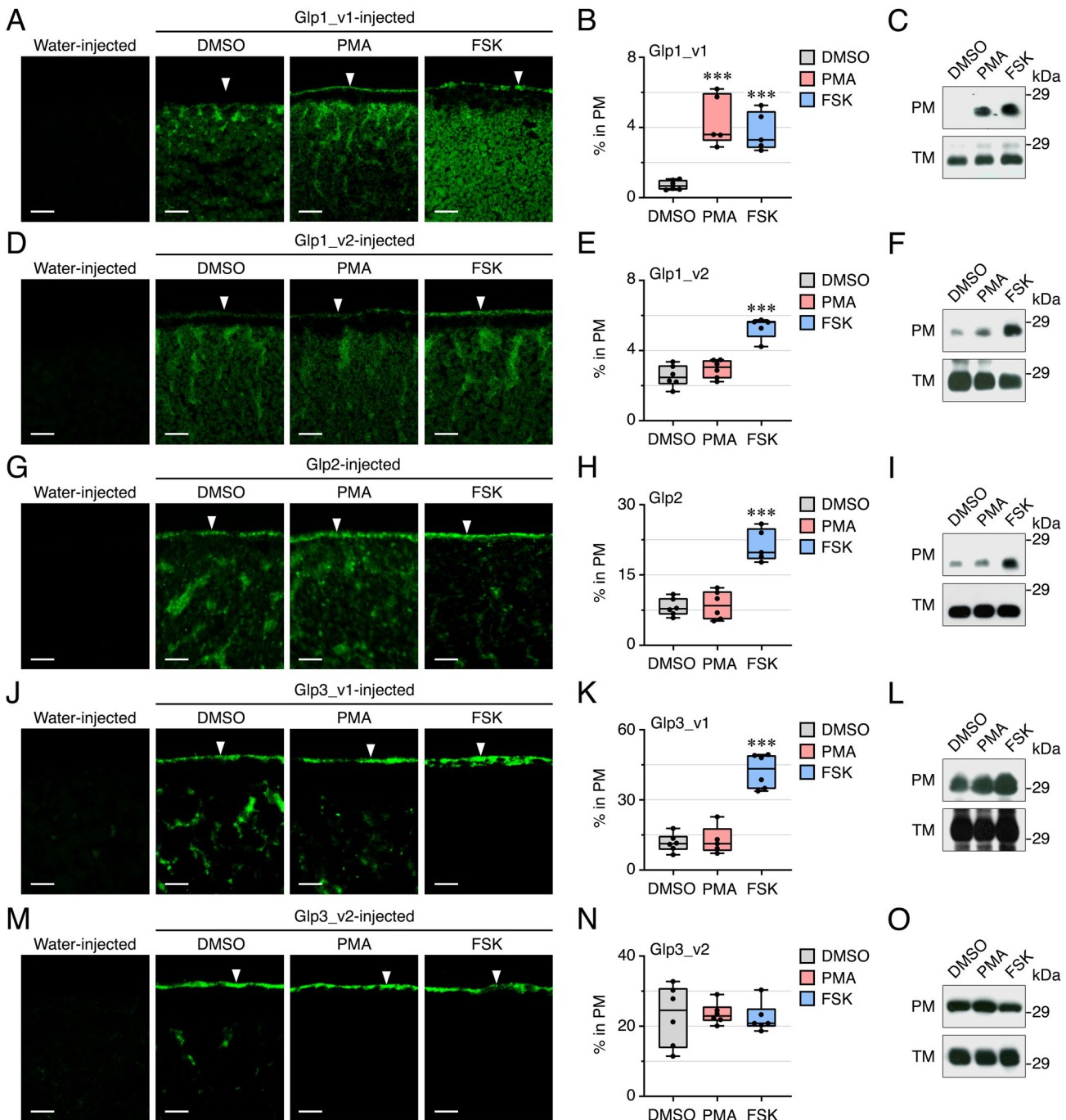

**Fig. 8 Regulation of *L. salmonis* Glp intracellullar trafficking in *X. laevis* oocytes. A**, **D**, **G**, **J**, **M** Representative immunofluorescence photomicrographs of paraffin sections of water and Glp-injected oocytes, the latter treated with the drug vehicle (DMSO, control), PMA or IBMX plus FSK. Scale bars, 25 μm. **B**, **E**, **H**, **K**, **N** Percentage of each Glp in the oocyte plasma membrane (PM) after each treatment determined by image analysis. Data are the mean ± SEM (*n* = 6 oocytes) were statistically analyzed by one-way ANOVA, followed by the Dunnett's multiple comparison test. \*\*\**p* < 0.001, with respect to DMSO-treated oocytes. **C**, **F**, **I**, **L**, **O** Representative immunoblots of Glp1_v1, Glp1_v2, Glp2, Glp3_v1 or Glp3_v2 in total membrane and plasma membrane purifications of oocytes expressing each Glp and treated with DMSO, PMA or IBMX/FSK. In all panels, the molecular mass markers (kDa) are on the right.

Glp3_v1) show differential subcellular localization. Since both of these paralogs are regulated by the PKA signaling pathway, other yet unknown mechanisms must also be involved in the intracellular trafficking regulation of the Glp1_v2 and Glp3_v1 channels.

In conclusion, we find that the evolution of Glps within the Crustacea is highly divergent, with large variances in gene copy numbers between the lineages. Species within the orders Pedunculata, Notostraca, Isopoda, Amphipoda and Decapoda typically retain single copy genes, while those within the orders Sessilia, Diplostraca, Euphausiacea, the class Copepoda, and the Palaemonidae family of decapod prawns have signifcantly expanded their *glp* gene repertoires. Gene expansion is associated with tandem duplications and bursts of transposon activities, rather than WGD. The highest copy numbers are currently found in the Daphnidae family of diplostracan water fleas, but the highest diversity is observed in Copepoda with large-scale genera-

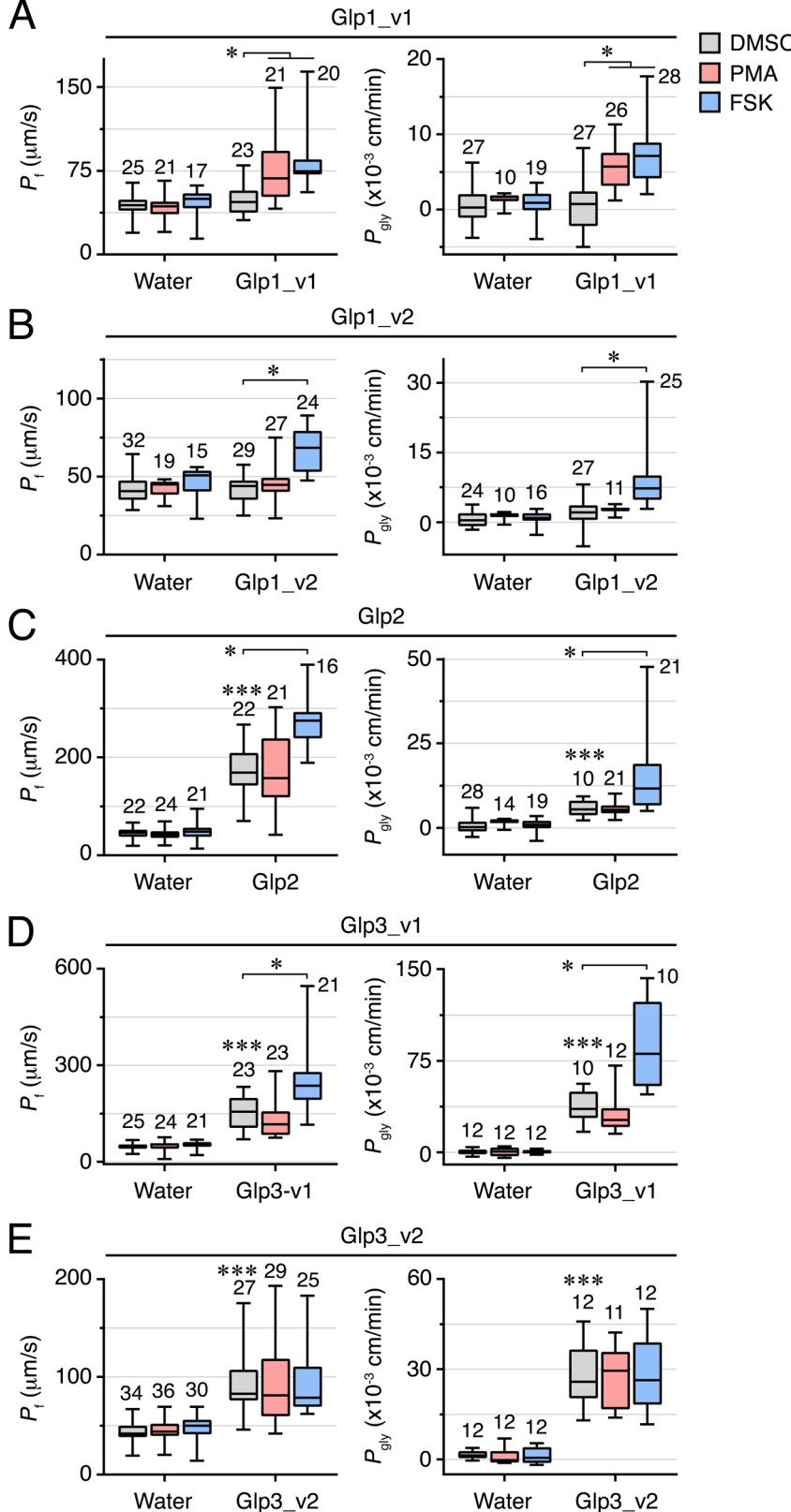

**Fig. 9 Function of *L. salmonis* Glps.** Osmotic water permeability ($P_f$; left panels) and glycerol permeability ($P_{gly}$) of frog oocytes injected with water (control) or Glp1_v1 (20 ng) (**A**), Glp1_v2 (20 ng) (**B**), Glp2 (0.25 ng) (**C**), Glp3_v1 (20 ng) (**D**) or Glp3_v2 (2 or 10 ng) (**E**) cRNAs. Before the swelling assays oocytes were exposed to DMSO, PMA or IBMX plus FSK. Data are the mean ± SEM (number of oocytes indicated on top of each bar) and were statistically analyzed by one-way ANOVA, followed by the Dunnett's multiple comparison test, or by the unpaired Student *t*-test. *$p < 0.05$; ***$p < 0.001$, with respect to non-treated controls or as indicated in brackets.

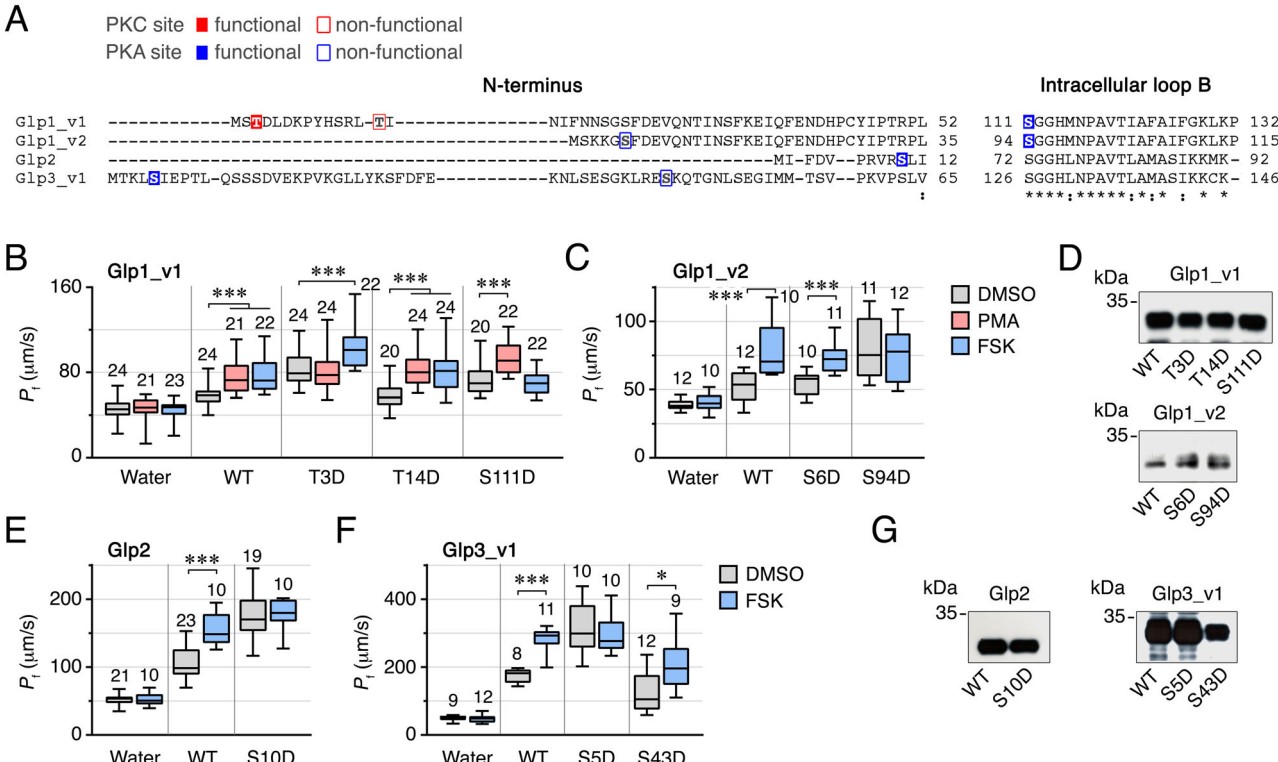

**Fig. 10 Identification of PKC and PKA phosphorylation sites in *L. salmonis* Glps. A** Amino acid alignment of the N-terminus and intracellular loop B of *L. salmonis* Glp1_v1, Glp1_v2, Glp2, Glp3_v1 and Glp3_v2. Asterisks and colons under the alignment indicate fully conserved residues and conservation between groups of strongly similar properties, respectively. Functional and non-functional phosphorylation sites by PKC (red) or PKA (blue) are indicated. **B**, **C**, **E**, **F** $P_f$ of oocytes injected with water or expressing wild-type Glps or mutant channels at the putative PKC and PKA phosphorylation residues and treated with PMA and/or IBMX/FSK. Data are the mean ± SEM (number of oocytes indicated on top of each bar). *$p < 0.05$; ***$p < 0.001$, statistically different (one-way ANOVA) as indicated in brackets. **D**, **G** Representative immunoblot of total membrane protein extracts from oocytes injected with each construct showing equivalent expression. Molecular mass markers (kDa) are on the left.

or species-specific duplications within two distantly related clusters. Based upon experimental evidence of the Glp proteins in a parasitic copepod, we find that *glp* gene duplication and splice variation has not resulted in functional redundancy of the channels. On the contrary, due to the evolution of unique regulatory sites for the PKA- and PKC-signal transduction pathways within the N-terminal or loop B domains of each paralog and isoform, the increased repertoire of Glps affords a high fidelity control over the channel membrane trafficking even when expressed in the same cell. These findings therefore suggest that neofunctionalization or subfunctionalization associated with intracellular trafficking represents an important selective force for Glp evolution in Copepoda.

## Methods

**Biological samples**. A laboratory strain of *L. salmonis* was raised on Atlantic salmon (*Salmon salar*)[47]. Prior to sampling of *L. salmonis* specimens, the fish were sedated with a mixture of benzocaine (60 mg/L) and methomidate (5 mg/L) and euthanised with a blow to the head. All experiments were conducted in accordance with the regulations approved by the governmental Norwegian Animal Research Authority (http://www.fdu.no/fdu/).

**Sequence, phylogenetic and syntenic analyses**. Contiguous peptide sequences were identified and assembled following tblastn queries of open source whole genome shotgun (WGS), transcriptome shotgun (TSA) and nucleotide databases (NCBI [blast.ncbi.nlm.nih.gov]). The corresponding nucleotide sequences were then retrieved from the respective databases and trimmed to match each peptide fragment, and finally concatenated to construct a coding sequence (CDS) for each gene or transcript[15,45]. Nucleotide sequence data reported are available in the Third Party Annotation Section of the DDBJ/ENA/GenBank databases under the accession numbers TPA: BK034896-BK035227. Prior to Bayesian (Mr Bayes

v3.2.2)[48] analyses, data sets of the deduced amino acids were aligned using the L-INS-I or G-INS-I algorithms of MAFFT v7.453[49], and converted to codon alignments using Pal2Nal[15,45,50]. Bayesian phylogenetic analyses with model parameters nucmodel = 4by4, nst = 2, rates = gamma were performed on the codon alignments following removal of the N- and C-termini and gapped regions containing less than three sequences. Two separate data sets of *glp* codons from Mandibulata (*N* = 241) and Copepoda (*N* = 98) were constructed (Supplementary Figs. 2 and 3) and analyzed with 15 and 5 million Markov chain Monte Carlo (MCMC) generations, respectively. Each run consisted of three heated and one cold chain with the resulting posterior distributions examined for convergence and an effective sample size >1400 using Tracer version 1.7[51]. Majority rule consensus trees were summarized with a burnin of 25%, processed with Archaeopteryx[52] and rendered with Geneious (Biomatters Ltd, New Zealand). Alignment files together with the accession numbers are provided in Supplementary data 1: Files S1-S4. The syntenic analyses of the *Daphnia* genes were conducted via tblastn searches of WGS databases. *In silico* searches for potential phosphorylation sites were carried out using the NetPhos 3.1 Server (http://www.cbs.dtu.dk/services/NetPhos/)[53].

***L. salmonis* Glp antibodies**. N-terminal or extracellular loop-C peptide sequences (Glp1_v1: MSTDLDKPYHSRLT; Glp1_v2: MSKKGSFD; Glp2: GYRSGPFVAG; Glp3_v1: KPVKGLLYKSFDFE; Glp3_v2: HSEGEGQNKDLEAT) were synthesized and injected in rabbits to raise paralog- and isoform-specific polyclonal antibodies (Agrisera AB, Sweden). The antisera were affinity purified against the synthetic peptides[54], and their specificity confirmed by ELISA, as well as by immunofluorescence microscopy and immunoblotting of *X. laevis* oocytes

**Functional characterization of *L. salmonis* Glps**. Constructs for heterologous expression in *X. laevis* oocytes were generated by subcloning the full-length *L. salmonis glp* cDNAs into pT7T expression vectors[23]. Point mutations in the wild-type sequences were introduced using the QuikChange Lightning Site-Directed Mutagenesis Kit (Agilent Technologies). All constructs in pT7T vectors were resequenced to validate that the correct mutations were present. The cRNA synthesis and isolation of stage V–VI oocytes were performed[55]. Oocytes were transferred to modified Barth's solution [MBS: 88 mM NaCl, 1 mM KCl, 2.4 mM

NaHCO$_3$, 0.82 mM MgSO$_4$, 0.33 mM Ca(NO$_3$)$_2$, 0.41 mM CaCl$_2$, 10 mM HEPES and 25 µg/ml gentamycin, pH 7.5] and injected with 50 nl of distilled water (control) or 50 nl of water containing 20 ng (Glp1_v1, Glp1_v2 and Glp3_v1), 2–10 ng (Glp3_v2) or 0.25 ng (Glp2) cRNA.

The $P_f$ of water-injected and Glp-expressing oocytes was determined using a swelling assay at pH 7.5[55,56]. The $P_{gly}$ was determined volumetrically in isotonic MBS at pH 7.5, where NaCl was replaced by 160 mM glycerol[57]. The osmolarity of the solution was measured with a vapor pressure osmometer (Vapro 5600, Wescor, USA), and adjusted to 200 mOsm with NaCl if necessary. The effect of the PKC activator PMA or the cAMP-PKA activator FSK on oocyte $P_f$ and $P_{gly}$ was respectively tested by preincubating the oocytes with 100 nM PMA for 30 min, or with 100 µM IBMX for 1 h and then with 100 µM FSK for 30 min, before the time-course determination of oocyte swelling.

**Protein extraction and immunoblotting**. Total and plasma membrane fractions of *X. laevis* oocytes were isolated[58]. *L. salmonis* specimens were dissociated with a glass dounce homogenizer in ice-cold RIPA buffer containing 150 mM NaCl, 50 mM Tris-HCl, pH 8, 1% Triton X-100, 0.5% sodium deoxycholate, 1 mM EDTA, 1 mM EGTA, EDTA-free protease inhibitors (Roche Applied Science, Mannheim, Germany), 1 mM Na$_3$VO$_4$ and 1 mM NaF, and centrifuged at 14,000 g for 10 min at 4 °C. The supernatant was mixed with 2× Laemmli sample buffer containing 200 µM dithiothreitol, heated at 95 °C for 15 min, deep frozen in liquid nitrogen, and stored at −80 °C. Protein extracts were deglycosylated with PNGase F (New England Biolabs) at 37 °C for 2 h and the reaction was blocked at 95 °C for 10 min before SDS-PAGE. Immunoblotted[45] membranes were incubated with Glp1_v1 (1:500), Glp1_v2 (1:500), Glp2 (1:500), Glp3_v1 (1:1000) or Glp3_v2 (1:1000) antibodies overnight at 4 °C. The specificity of the reactions was determined by incubation of duplicate membranes with the antiserum preadsorbed with the corresponding antigenic peptide.

**Immunofluorescence microscopy**. Fixation of *X. laevis* oocytes and *L. salmonis* specimens, and processing for immunostaining on histological sections[55]. Sections were incubated with 1:300 dilutions for each primary antibody and 1.1000 dilution of Cy3-conjugated anti-rabbit antibody. Labeled sections were photographed at 63x magnification with a Zeiss Axio Imager Z1/ApoTome fluorescence microscope (Carl Zeiss Corp., Belcodène, France). Images from negative control sections were taken with the same fluorescence intensity and exposure times than those used for the positives. In *X. laevis* oocytes, the relative abundance of each Glp at the oocyte surface, in the presence or absence of PMA and FSK, was semiquantified using the ImageJ open-source software (version 1.46r). A section of the oocyte with fixed dimensions enclosing the plasma membrane and cytoplasm was generated, and the pixel intensity within each region was recorded. The dimensions of the oocyte section were kept constant for all images from oocytes expressing the same Glp. The pixel values from one image from six oocytes per treatment were analyzed.

**Statistics and reproducibility**. Data (mean ± SEM) on the percentage of Glp in the oocyte plasma membrane, $P_f$ and $P_{gly}$ were statistically analyzed by one-way ANOVA, followed by the Dunnett's multiple comparison tests, or by an unpaired Student's $t$ tests. Data were tested for normal distribution (Shapiro-Wilk test) and homogeneity of variances (Forsythe and Barlett test) prior to conducting parametric tests. Statistical analyses were carried out using the SigmaPlot software v12.0 (Systat Software Inc.) and GraphPad Prism v8.4.3 (686) (GraphPad Software). In all cases, statistical significance was defined as $P < 0.05$ (*), $P < 0.01$ (**), or $P < 0.001$ (***).

**Reporting summary**. Further information on research design is available in the Nature Research Reporting Summary linked to this article.

## Data availability

All data generated or analyzed during this study are included in this manuscript (and its supplementary information files). The complete alignments shown in Fig. 1 and Supplementary Fig. 1 are provided in Supplementary Data 1. The source data underlying plots shown in figures are provided in Supplementary Data 2. All other data is available from authors upon reasonable request.

## Code availability

Mr Bayes software used for phylogenetic inference is available at https://github.com/NBISweden/MrBayes/releases. MAFFT used for multiple sequence alignments is available at https://mafft.cbrc.jp/alignment/software. PAL2NAL used for codon conversion is available at http://www.bork.embl.de/pal2nal/.

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

## Acknowledgements

This work was supported by grants from the Norwegian Research Council (Grants no. 254872/E40, 204813/E40 and 294768/E40 to R.N.F., and 203516/O30 to F.N.) and the Spanish Ministry of Science and Innovation (MICINN) (Grant no. PID2019-103868RB-I00 to J.C.). Participations of M.C. and J.A.S. were funded by the Norwegian Research Council (Grant nos 254872/E40 and 204813/E40, respectively). Participation of F.C. was funded by a "Ramon y Cajal" contract (RYC-2015-17103) from the Spanish MINECO.

## Author contributions

R.N.F., M.C. and J.C. designed the research; M.C., F.C., J.A.S. and R.N.F. performed the research; F.N. contributed with materials and the acquisition of animal samples; M.C., R.N.F. and J.C. analyzed the data and M.C., R.N.F. and J.C. wrote the manuscript.

## Competing interests

The authors declare no competing interests.
