## [Peer Review File · Communications Biology]

Reviewers' comments:

Reviewer #1 (Remarks to the Author):

In the Manuscript 'Lineage-level divergence of copepod 1 glycerol transporters and the emergence of isoform-specific trafficking regulation' and submitted to Comm Biol for consideration, the authors focused upon gene diversity of aquaglyceroporins (GLPs) as well as on the functional role of these proteins in Crustacea. With this purpose, they used different species (pedunculate barnacles, tadpole shrimps, isopods, amphipods...) and found that while GLPs were present as single copy genes in some species, others showed up to 10 copies or high rates of gene duplication, which was related to mono/polyphyletic lineage evolution. These experiments were followed up by others of functional nature in which gene duplication and splicing were related to new functions (paralogs). This second part of the work, which was entirely conducted on *Lepeophtheirus salmonis*, showed spliced variants related to specific tissue and sex expression of GLPs and to PKC- or PKA-dependent mechanisms (Glp1_v1, Glp1_v2 Glp2 and Glp3_v1). The authors conclude that the apparent relationship of mutations in GLP genes with precise regulation of intracellular tracking may be related to the evolution of gene copies and neofunctionalization in these species.

GENERAL COMMENTS

This is an interesting study that comes to fill a niche in our understanding about how gene duplications and neofunctionalization has evolved in Crustacea. Objectives are clear, the use of the literature is adequate, experiments have been well devised, methods are sound, and conclusions align well with the authors' data. I don't have major concerns on the scientific approach and content, but just some specific comments that may be taken into account while the authors revise their paper.

SPECIFIC COMMENTS

Please check that all gene names are italicized and that notation to proteins and genes is adequate and allows the readers to clearly see whether the authors are referring to genes or proteins

Abstract

L35 'species-specific'

Introduction

L56 'In plants and insects, (...)'

L62 'organisms such as *Drosophila* has thus obfuscated...'

L79 'which expresses glp1- and...'

Results

Ensure that all this section is written in past tense when referring to the results obtained in this work

L122, L124 'were found'

L125 'were encoded'

L136 'myriapods indicates...'

L151 'revealed'

L155 'analysis was'

L171 'showed'

L216 'expressed in enterocytes,'

Discussion

L317 'whereas diverse lineages...'

L403 'mechanisms'

L406 'we found'

M&M

L433 'experiments'

L444 '(Mr Bayes v3.2.2) analyses'

L498 space between figures and units

L526 'one image from six oocytes'

L529 Please state that data were tested for normal distribution (Shapiro-Wilk test) and homogeneity of variances (Levene or Barlett tests) prior to conducting parametric tests (i.e. ANOVA and t-test).

Reviewer #2 (Remarks to the Author):

The manuscript is about aquaglyceroporins evolution in crustacean, transporter neofunctionalization with tissue or sex-specific expression of the proteins, and also with sites for protein kinase C (PKC)- or protein kinase A (PKA)- regulation of intracellular membrane trafficking.

The experimental analyses are very sophisticated, and the authors make a good use of the available tools to answer the investigate questions. The obtained results present a good contribution for the knowledge of evolution and functioning of invertebrate transport proteins.

I have just few minor points to indicate,

. Abstract, line 27 - I suggest replacing the word facilitated, it can induce the type of transport "facilitated diffusion" carried out by carrier proteins, which is not the case, but a channel protein
. keywords are not welcome?

. In the introduction, I think that for reading to flow easier there could be a better connection between the phylogenetic and functional approach. In order to prepare the reader, especially for the results of cell localization and regulation of different protein isoforms

. In the results, references to figures of the supplementary material appear, I did not find such figures (only their captions)

. In some moments of the results the authors say "... that the results SUGGEST ...", be careful, this indicates interpretation, which does not fit in results, but in the discussion (for example, lines 217, 255...). And the last sentence of the Results seems like a conclusion, which also doesn't fit in this section.

Considering that the manuscript deals with evolutionary aspects related to the functionality of aquaglyceroporins, it would be relevant to contextualize its function in different tissues of crustaceans, both in the introduction and in the discussion.

Reviewer #3 (Remarks to the Author):

I found this manuscript to be well-written. The evolution of Glps, especially in Crustacea is something that has yet been covered in full details, making this paper that much more original. It was interesting to see that unlike most taxa of crustacea, copepods (*L. salmonis*) show high glp redundancy along with isoform specific pathway regulation. I agree with the author on most aspects covered in this manuscript, however I have some major concerns. While the results are somewhat complete, I believe the discussion needs major improvement for the authors make conclusion they have made in this MS. I am unfortunately giving this paper a rejection, but please refer to both the comments here, as well as ones left in the reviewer's copy to improve on this manuscript for future resubmission.

Concerns:

1. Line 357-359, authors state that they have found evidence that functional repertoires are further increased through splice variation, but I've failed to find any figures, tables or mention within the results which allude back to this statement. Please clarify or elaborate on this further.
2. Authors assert that copepod glps are evolving at a high rate, but are also positively selected for specific purpose (line 358-360). While I do not disagree with this statement with the given results, to make this statement, I believe the authors need a statistical test which tests significance of positive selection in copepods (via tools such as CAFE, Computational Analysis of Gene Family Evolution).

3. The authors conclude the paper by suggesting that neofunctionalization associated with intracellular trafficking represents an important selective force for Glp evolution in Copepoda. While this statement can be true, perhaps it would be good to list out all other possible selective forces (eg. Subfunctionalization) that could be at play (and ruling them out), in the discussion, since it is yet premature to make absolute proclamations (especially with copepods being unique in having way more glp paralogs compared to most other crustacea). Adding to this, discussion and conclusion in which the authors have arrived at feels premature, meaning much more in-depth exploration of the discussion is still required.

Basic reporting

Aside from simple typos, grammatical errors and form inconsistencies, the general flow of the manuscript is well written. Also, the term "adsorption" and "absorption" seem to be used interchangeably within the manuscript (whether intended or not), please double check this aspect of the manuscript.

Abstract is on point and concise.

Introduction is well-structured, which highlights all necessary explanation such as introduction to glps, importance of exploring glp evolution in crustacea as well as previous studies.

Methods are in-depth and clear.

Results are complete.

Discussion and conclusion require much more in-depth exploration.

Reference list is complete.

Response to reviewers

We have proofread the manuscript for typos, improved the flow of the text, qualified the discussion, and addressed each of the points raised by the reviewers as detailed below:

Reviewer #1

In the Manuscript ‘Lineage-level divergence of copepod 1 glycerol transporters and the emergence of isoform-specific trafficking regulation’ and submitted to Comm Biol for consideration, the authors focused upon gene diversity of aquaglyceroporins (GLPs) as well as on the functional role of these proteins in Crustacea. With this purpose, they used different species (pedunculate barnacles, tadpole shrimps, isopods, amphipods...) and found that while GLPs were present as single copy genes in some species, others showed up to 10 copies or high rates of gene duplication, which was related to mono/polyphyletic lineage evolution. These experiments were followed up by others of functional nature in which gene duplication and splicing were related to new functions (paralogs). This second part of the work, which was entirely conducted on *Lepeophtheirus salmonis*, showed spliced variants related to specific tissue and sex expression of GLPs and to PKC- or PKA-dependent mechanisms (Glp1_v1, Glp1_v2 Glp2 and Glp3_v1). The authors conclude that the apparent relationship of mutations in GLP genes with precise regulation of intracellular tracking may be related to the evolution of gene copies and neofunctionalization in these species.

GENERAL COMMENTS

This is an interesting study that comes to fill a niche in our understanding about how gene duplications and neofunctionalization has evolved in Crustacea. Objectives are clear, the use of the literature is adequate, experiments have been well devised, methods are sound, and conclusions align well with the authors’ data. I don’t have major concerns on the scientific approach and content, but just some specific comments that may be taken into account while the authors revise their paper.

SPECIFIC COMMENTS

Please check that all gene names all italicized and that notation to proteins and genes is adequate and allows the readers to clearly on whether the authors are referring to genes or proteins

1. Abstract

L35 ‘species-specific’
- corrected (see line 35)

2. Introduction

L56 ‘In plants and insects, (...)’
- corrected (see line 56)

3. L62 ‘organisms such as *Drosophila* has thus obfuscated...’
 - corrected (see line 61)
4. L79 ‘which expresses *glp1*- and...’
 - the use of the definite article “the” is necessary in the sentence “which expresses the *glp1* and -3 genes”, and is therefore not removed (see line 78).
5. Results
 - Ensure that all this section is written in past tense when referring to the results obtained in this work
 - L122, L124 ‘were found’
 - L125 ‘were encoded’
 - L136 ‘myriapods indicates...’
 - L151 ‘revealed’
 - L155 ‘analysis was’
 - L171 ‘showed’
 - L216 ‘expressed in enterocytes,’
 - These sentences were not corrected as the Communications Biology style and formatting guide specifically requires the major results and conclusions of the paper in the present tense.
6. Discussion
 - L317 ‘whereas diverse lineages...’
 - corrected (see line 317)
7. L403 ‘mechanisms’
 - corrected (see line 402)
8. L406 ‘we found’
 - not corrected
- 9- M&M
 - L433 ‘experiments’
 - corrected (see line 430)
10. L444 ‘(Mr Bayes v3.2.2) analyses’
 - corrected (see line 440)
11. L498 space between figures and units
 - corrected (see line 494)
12. L526 ‘one image from six oocytes’
 - corrected (see line 520-521)

13. L529 Please state that data were tested for normal distribution (Shapiro-Wilk test) and homogeneity of variances (Levene or Barlett tests) prior to conducting parametric tests (i.e. ANOVA and t-test).

- corrected (see lines 526-528)

Reviewer #2

The manuscript is about aquaglyceroporins evolution in crustacean, transporter neofunctionalization with tissue or sex-specific expression of the proteins, and also with sites for protein kinase C (PKC)- or protein kinase A (PKA)- regulation of intracellular membrane trafficking.

The experimental analyses are very sophisticated, and the authors make a good use of the available tools to answer the investigate questions. The obtained results present a good contribution for the knowledge of evolution and functioning of invertebrate transport proteins.

I have just few minor points to indicate,

1. Abstract, line 27 - I suggest replacing the word facilitated, it can induce the type of transport "facilitated diffusion" carried out by carrier proteins, which is not the case, but a channel protein
- corrected (see line 27)
2. keywords are not welcome?
- keywords are provided in the submission form
3. In the introduction, I think that for reading to flow easier there could be a better connection between the phylogenetic and functional approach. In order to prepare the reader, especially for the results of cell localization and regulation of different protein isoforms
- we have added a sentence in the introduction to improve the flow of the text between the phylogenetic and functional studies (see lines 94-95)
4. In the results, references to figures of the supplementary material appear, I did not find such figures (only their captions)
- all data, including the Supplementary Figures and Supplementary files were uploaded and available on the submission servers
5. In some moments of the results the authors say "... that the results SUGGEST ...", be careful, this indicates interpretation, which does not fit in results, but in the discussion (for example, lines 217, 255...). And the last sentence of the Results seems like a conclusion, which also doesn't fit in this section.
- corrected (see lines 217 and 255)

6. Considering that the manuscript deals with evolutionary aspects related to the functionality of aquaglyceroporins, it would be relevant to contextualize its function in different tissues of crustaceans, both in the introduction and in the discussion.
- we have added a sentence in the introduction to contextualize the functionality of aquaglyceroporins (see lines 94-95), and modified an explanatory sentence in the discussion to contextualize the Glp function in the different tissues (see lines 358-360).

Reviewer #3

1. Line 357-359, authors state that they have found evidence that functional repertoires are further increased through splice variation, but I've failed to find any figures, tables or mention within the results which allude back to this statement. Please clarify or elaborate on this further.
- we have added a sentence in the results with reference to Fig. 1C and Supplementary Fig. 1A to clarify this (see lines 156-157).
2. Authors assert that copepod glps are evolving at a high rate, but are also positively selected for specific purpose (line 358-360). While I do not disagree with this statement with the given results, to make this statement, I believe the authors need a statistical test which tests significance of positive selection in copepods (via tools such as CAFE, Computational Analysis of Gene Family Evolution).
- the statement was based upon the observation of broad-levels of species- and genera-specific gene duplications as well as of splice variants in three of the orders. We subsequently experimentally demonstrated the divergent tissue expression and trafficking regulation of the derived proteins. In this context, we believe that a predictive statistic is not necessary. However, to avoid terminological ambiguity we have now removed that sentence (see lines 356-357).
3. The authors conclude the paper by suggesting that neofunctionalization associated with intracellular trafficking represents an important selective force for Glp evolution in Copepoda. While this statement can be true, perhaps it would be good to list out all other possible selective forces (eg. Subfunctionalization) that could be at play (and ruling them out), in the discussion, since it is yet premature to make absolute proclamations (especially with copepods being unique in having way more glp paralogs compared to most other crustacea). Adding to this, discussion and conclusion in which the authors have arrived at feels premature, meaning much more in-depth exploration of the discussion is still required.
- we have modified a sentence in the discussion (see line 368) and conclusion (see line 420), and added a sentence (see lines 374-376) in the discussion to clarify the potential roles of neofunctionalisation and subfunctionalisation.
4. Basic reporting
Aside from simple typos, grammatical errors and form inconsistencies, the general flow of the manuscript is well written. Also, the term "adsorption" and "absorption"

seem to be used interchangeably within the manuscript (whether intended or not), please double check this aspect of the manuscript.

- corrected to preadsorbed throughout the manuscript

Abstract is on point and concise.

Introduction is well-structured, which highlights all necessary explanation such as introduction to glps, importance of exploring glp evolution in crustacea as well as previous studies.

Methods are in-depth and clear.

Results are complete.

Discussion and conclusion require much more in-depth exploration.

Reference list is complete.

We thank the reviewers for their constructive comments

REVIEWERS' COMMENTS:

Reviewer #2 (Remarks to the Author):

I am satisfied with the review of the text by the authors.

Reviewer #3 (Remarks to the Author):

All questions are clear!